# Effects of Deformed Wing Virus-Targeting dsRNA on Viral Loads in Bees Parasitised and Non-Parasitised by *Varroa destructor*

**DOI:** 10.3390/v15112259

**Published:** 2023-11-15

**Authors:** Zoe E. Smeele, James W. Baty, Philip J. Lester

**Affiliations:** School of Biological Sciences, Victoria University of Wellington, P.O. Box 600, Wellington 6140, New Zealand; james.baty@vuw.ac.nz (J.W.B.); phil.lester@vuw.ac.nz (P.J.L.)

**Keywords:** deformed wing virus, RNA interference, double-stranded RNA, varroa destructor, biopesticide, RNA-seq

## Abstract

The *Varroa destructor* mite is a devastating parasite of honey bees; however the negative effects of varroa parasitism are exacerbated by its role as an efficient vector of the honey bee pathogen, Deformed wing virus (DWV). While no direct treatment for DWV infection is available for beekeepers to use on their hives, RNA interference (RNAi) has been widely explored as a possible biopesticide approach for a range of pests and pathogens. This study tested the effectiveness of three DWV-specific dsRNA sequences to lower DWV loads and symptoms in honey bees reared from larvae in laboratory mini-hives containing bees and varroa. The effects of DWV-dsRNA treatment on bees parasitised and non-parasitised by varroa mites during development were investigated. Additionally, the impact of DWV-dsRNA on viral loads and gene expression in brood-parasitising mites was assessed using RNA-sequencing. Bees parasitised during development had significantly higher DWV levels compared to non-parasitised bees. However, DWV-dsRNA did not significantly reduce DWV loads or symptoms in mini-hive reared bees, possibly due to sequence divergence between the DWV variants present in bees and varroa and the specific DWV-dsRNA sequences used. Varroa mites from DWV-dsRNA treated mini-hives did not show evidence of an elevated RNAi response or significant difference in DWV levels. Overall, our findings show that RNAi is not always successful, and multiple factors including pathogen diversity and transmission route may impact its efficiency.

## 1. Introduction

Honey bees (*Apis mellifera*) can be infected with a wide range of viruses that can cause severe disease in colonies. Several of these viruses can be vectored by the ectoparasite *Varroa destructor* while it feeds on the honey bee’s body fat [1,2,3,4,5,6]. However, *V. destructor* (varroa hereafter) parasitism is most closely associated with Deformed wing virus (DWV), and together they represent the most pressing health concern to honey bee populations around the world [7,8,9,10,11,12].

The global spread of varroa has rapidly transformed DWV into a pandemic among honey bee populations [13], dramatically increasing the prevalence and loads of DWV in infested colonies [8,14,15]. Deformed wing virus can cause wing and abdominal deformities in developing pupae, a phenotype highly associated with varroa parasitism [1,16]. However, parasitised bees that do not develop this deformed phenotype can still have high loads of DWV [16,17] and exhibit behavioural symptoms. For example, DWV infection has been shown to impair associative learning and memory [18], reduce the life span of individuals, decrease homing ability and lead to precocious foraging and behavioural maturation [19,20]. These multifaceted, adverse effects of DWV infection have large implications on the overall health of the colony by devastating the working force that maintain the hive.

Currently, there is no direct control method for DWV. Studies have shown that keeping varroa levels low in a colony through acaricide treatments can keep colony-level loads of DWV low compared to untreated colonies [21,22]. Beekeepers typically treat their colonies for varroa at least twice a year, often during spring and autumn. However, delayed autumn treatments can significantly affect over-winter hive survival due to DWV infection in the population of bees that overwinter in the hive, known as winter bees [21,22]. With no turnover in the bee population over winter, an infection of DWV among winter bees is not cleared from the colony and its negative effects are perpetuated, resulting in significantly weakened colonies. Moreover, the indirect control of DWV relies on effective acaricide treatments, and reports of resistance to treatments among some mite populations is increasing [23,24,25]. Overall, the development of more effective and sustainable control strategies for varroa and DWV is essential for the apiculture industry globally.

RNA interference (RNAi) has been widely proposed as a targeted and sustainable pest-control strategy, particularly in agriculture [26,27,28,29] as well as conservation [30,31]. RNA interference is an intracellular mechanism of sequence-specific gene silencing conserved across eukaryotes. There are four main RNAi pathways that differ based on the type of RNA molecule that triggers them. In invertebrates and plants, the major pathway involved in antiviral immunity is the small interfering RNA (siRNA) pathway. This pathway is triggered via the recognition of double-stranded RNA (dsRNA) molecules in the cell cytosol [32]. The foreign dsRNA is cleaved into 21–22 bp siRNA by an RNase-III Dicer enzyme, and one of these siRNA strands is then bound to the endoribonuclease Argonaute, which incorporates it into the multiprotein RNA-induced silencing complex (RISC; [32]). Bound to RISC, the guide strand is used as a sequence-specific template to target complementary messenger RNA (mRNA) sequences or viral RNA for cleavage [32]. RNAi-based technologies for pathogen or pest control exploit this pathway to suppress the expression of specific transcripts through the delivery of sequence-specific dsRNA or siRNA complementary to mRNA transcripts that encode for proteins important to the survival or reproduction/replication of the target organism or entity.

Over the last decade, research has explored the efficacy of using RNAi to control several common honey bee pathogens and parasites including, varroa [33,34,35], *Nosema ceranae* [36,37], Israeli acute paralysis virus (IAPV) [38,39], Sacbrood virus (SBV) [40,41] and the small hive beetle (*Aethina tumida*) [42]. Many of these studies have shown that oral administration of pathogen-specific dsRNA to honey bees can lead to reduced viral loads and disease symptoms within bees [38,40,43,44]. Specifically, two studies have tested the use of RNAi to reduce the development of DWV infection in honey bees with promising results. Desai et al. (2012) showed that feeding DWV-inoculated adults and pupae with DWV-specific dsRNA could significantly reduce the development of DWV infections compared to control bees fed with either sugar water or green fluorescent protein (GFP)-specific dsRNA. More recently, Leonard et al. (2020) tested the effects of DWV-specific dsRNA on DWV loads and bee survival by inoculating newly emerged honey bees with a gut symbiont that was genetically-modified to express DWV-specific dsRNA. Newly emerged bees experimentally infected with DWV developed significantly reduced infections when inoculated with gut symbionts expressing DWV-specific dsRNA compared to virus infected control bees [35]. These DWV RNAi studies have shown promising results for the effectiveness of using DWV-specific dsRNA as a prophylactic treatment of DWV to protect bees against developing high infection levels. However, most RNAi studies have not tested the efficiency of pathogen-specific dsRNA to reduce levels of naturally acquired virus infections. Such studies are important for understanding the effects of RNAi on viral loads that would normally be observed within a hive. Targeting DWV in developing larvae (i.e., early in infection development) may be highly important for mitigating many negative colony-level effects of the virus. Additionally, RNAi studies have neglected to test the effects of pathogen-specific dsRNA under varroa parasitism conditions, which represents a highly important transmission route for many honey bee pathogens, especially DWV.

Our goal in this study was to build upon previous DWV-RNAi studies by testing the effectiveness of feeding a cocktail of three DWV-specific dsRNA (DWV-dsRNA) sequences to lower DWV loads and symptoms in reared honey bees. The experiment employed mini-hives within a laboratory to test the effects of DWV-dsRNA treatment under conditions closely resembling that of a colony, and therefore, mini-hives of bees were not inoculated with DWV. These mini-hives were experimentally infested with varroa mites to test the effectiveness of DWV-dsRNA treatment in bees parasitised and non-parasitised by varroa during their development. The three DWV-dsRNA sequences used in this study have previously been shown to significantly knock down DWV loads in adults and larvae [35,43]. Mini-hives have been used in previous RNAi research to show that dsRNA fed to adult bees can be transmitted to and knockdown gene expression in larvae [45] and parasitising mites [34]. By resembling the within-hive dynamics, mini-hives represent a more realistic scenario for assessing treatment effects while maintaining the controlled conditions of being conducted in the lab. In our study, young bees were reared in mini-hives from larvae, and sampled by uncapping brood cells when they were close to emerging. Deformed wing virus levels in uncapped bees were measured using quantitative PCR (qPCR), and wing deformities were recorded to assess differences in DWV symptoms. We investigated the effects of DWV-dsRNA treatment on viral loads and gene expression in brood-parasitising foundress varroa mites using RNA-sequencing (RNA-seq) of pooled foundress mites. We expected parasitised bees to have elevated DWV loads compared to non-parasitised bees irrespective of treatment, consistent with the known effects of varroa parasitism [16]. However, we hypothesized that DWV loads would be significantly lower in bees and varroa from DWV-dsRNA treated mini-hives compared to mini-hives treated with a non-specific dsRNA or sugar water controls.

## 2. Materials and Methods

### 2.1. Mini-Hive and Mini-Frame Design

Mini-frames were constructed to a dimension of 23 cm by 21.5 cm with 1.5 cm overhangs along the top of frames. One sheet of wax foundation was melted into the inside of the frame, and mini-frames were adhered together using tape (Figure 1a). When adhered side by side, mini-frames fit in a hive box as one full depth frame. Mini-frames were provided to colonies prior to the start of the experiment for wax foundation to be drawn out as comb. Hive Doctor Smart Nuc^®^ full-depth 5-frame nucleus boxes (Ecrotek, Auckland, New Zealand) were assembled with attachments per manufacturer’s instructions and used as mini-hive boxes with the following modifications (Figure 1b). Two 20 cm × 20 cm plexiglass viewing windows were tapped into the front side of the box using fabric tape. A plexiglass divider (23 cm × 20 cm) was secured in the middle of the box to separate a brood and foraging chamber and stabilise the mini-frame when inserted into the brood chamber. A gap between the top of the divider and the lid of the box allowed bees to move between chambers. A flap in the back of the foraging chamber was made to insert a dsRNA treatment pouch. Additionally, a 20 cm × 16 cm area was cut out of the mini-hive lid above the foraging chamber and replaced with 1 mm^2^ metal mesh by hot gluing the mesh over the cut out square. This mesh lid over the foraging chamber provided extra ventilation and was used to provide bees with water and the pollen patties, as described below.

### 2.2. Maintaining Laboratory Mini-Hives

Three honey bee colonies from the Victoria University of Wellington, New Zealand campus were used as donor colonies of adult honey bees and brood for the experiment. Varroa were collected through sugar shake or uncapping of drone brood from highly infested hives from a local Wellington beekeeper. Varroa collected from the infested hives were likely actively vectoring DWV given the presence of many bees displaying wing deformities. Bees, brood and varroa were all collected for the experiment on the same day.

In order to obtain mini-frames of similarly aged larvae, Ceracell™ Nicot Queen Introduction Cages (Auckland, New Zealand) were used in donor colonies to trap the queen and 10–15 attendant workers to one side of a mini-frame for 24 h. Cages were stabilised onto the comb using two rubber bands. Mini-frames were monitored daily for when larvae reached approximately second instar. Mini-frames were then removed from the donor hive and introduced to the mini-hive with approximately 300 nurse bees. Mini-frames were placed with the side containing the monitored larvae facing the plexiglass window. A mini-frame of empty comb or wax foundation was also placed in the mini-hive brood chamber to fill excess space. Each mini-hive was experimentally infested with 50–60 varroa mites that were collected on the same day. Prior to bees and brood being introduced into the mini-hive, mites were counted onto a damp paper towel and placed on the bottom of the mini-hive brood chamber, directly below where the brood mini-frame would be inserted (Appendix A). Only actively moving varroa mites were used in the experiment. Mini-hive replicates were temporally separated with weekly cohorts of each treatment brought into the lab. In total, each treatment contained four mini-hive replicates, except the DWV-dsRNA treatment where a fifth replicate was run. Bees, brood and varroa were collected and brought into the lab in mini-hives during austral late summer to early autumn, early February to mid-March 2022.

One tube lure of queen mandibular pheromone (TempQueen with QDP) was provided to each mini-hive and placed in the brood chamber to mimic the presence of a queen. To meet their nutritional requirements as advised in the COLOSS BEEBOOK [46], each mini-hive was provided with a square of bee bread (~9 cm × 5 cm) cut out from a frame of stored pollen (Appendix A). A protein/honey supplement paste was also provided to each mini-hive in the form a 10:1 MegaBee© patty (MegaBee, Tucson, AZ, USA) and honey paste mixture. This paste was spread over the mesh top on the foraging chamber and reapplied as the bees finished it (which was almost every day prior to cell capping). Tap water was provided to mini-hives ad libitum in the form of filled 50 mL plastic collection jars with holes drilled in the caps, placed upside down on top of a cotton ball pad. One jar/cotton ball was placed on the mesh top of the foraging chamber of each mini-hive and replenished daily. Mini-hives were maintained in a temperature-controlled room on a full dark–light cycle at 31–33 °C and approximately 50% relative humidity.

### 2.3. Double-Stranded RNA Treatments

Mini-hives were fed one of three treatments as their main sugar source. This was an 80% sucrose solution containing either: 0.06 mg/mL of DWV-targeting dsRNA cocktail (DWV-dsRNA), 2 mg/mL of a dsRNA non-specific to bees, varroa or associated pathogens (non-specific dsRNA) or a dsRNA-free sugar-water control consisting of only 80% sucrose (Sugar control). Non-specific dsRNA and sugar water control treatments were run as controls alongside a concurrent RNAi experiment. The non-specific dsRNA matched a sequence of the MPK4a soybean (*Glycine max*) gene without complementary sequences in honey bees or varroa mites. The DWV-dsRNA sequences were synthesised using RNA Greentech (Texas, USA) with magnetic-bead precipitation. Three non-structural encoding regions of the DWV genome [35], specifically 3Cprotease, helicase and RNA-dependent RNA polymerase regions, were used as templates to synthesize three dsRNA sequences. Complete sequences for each DWV-dsRNA synthesized are provided in Appendix A. Sequences arrived in a stock concentration of 1.6–1.8 mg/mL and were combined with 80% sucrose solution to a final concentration of 0.06 mg/mL.

Double-stranded RNA treatments were provided in a pouch with perforated holes made on the upward facing side for bees to feed from and placed on the bottom of the mini-hive foraging chamber. DWV-dsRNA treated mini-hives were provided with 30 mL of DWV-dsRNA (1.8 mg of dsRNA) each day until all cells were capped (five days). Each day bees were observed while feeding, and they consumed all 30 mL of DWV-dsRNA treatment. Mini-hives were then switched to 80% sugar water to maintain adult bees until the end of the experiment. Non-specific dsRNA and sugar control-treated mini-hives were each provided a pouch containing 500 mL of treatment which was sufficient for the duration of the experiment.

### 2.4. Experiment Sampling

Mini-frames of the brood were monitored daily through the mini-hive window to assess cell capping and the time until adult bees began emerging (Appendix A). Mini-hives were removed from the experiment either 12 days after cells were capped or when young bees began chewing their cell capping (approximately 16 days after bringing a mini-hive of bees into the lab). Mini-hives were first anesthetized with CO_2_ and the frame of the capped brood was removed. All cells in the brood frame were then individually uncapped under a Nikon SMZ645 stereo microscope by peeling off the wax capping with forceps. Young bees were removed, and we recorded their age, survival, parasitism status and DWV symptoms (normal or deformed wings). Bees were aged based on morphological characteristics [47]. All brood-parasitising foundress mites were collected from uncapped cells. All uncapped bees were separated into three parasitism phenotypes: parasitised with normal wings; parasitised with deformed wings; and non-parasitised. All parasitised bees and a subsample of 30 non-parasitised bees were collected from each mini-hive. All samples were stored at −80 °C prior to molecular analysis.

### 2.5. Prevalence of Wing Deformities in Uncapped Bees

We analysed the proportion of bees showing wing deformities using a generalized linear model fit with a quasibinomial distribution (to account for over dispersion) to investigate treatment effects on wing deformities among uncapped bees. The model was tested with a type 2 ANOVA to determine if treatment significantly explained differences in the prevalence of wing deformities in uncapped bees.

### 2.6. Relative Quantification of Deformed Wing Virus in Uncapped Bees

To assess the effect of DWV-dsRNA treatment on DWV loads in uncapped bees, total RNA was extracted from six individual uncapped bees from each mini-hive (two bees per parasitism phenotype where possible). From one sugar water treated mini-hive, only parasitised bees were sampled for RNA extraction. Bees were individually placed in 2 mL reinforced tubes (Sarstedt, Germany) with two 3.2 mm stainless steel beads, 1 mL of GENEzol™ DNA Reagent Plant (Geneaid Biotech Ltd., New Taipei City, Taiwan) and 5 µL of beta-mercaptoethanol (0.5%) (Sigma-Aldrich, St. Louis, MI, USA). Samples were then homogenised using a Precellys^®^ Evolution homogeniser (Bertin Instruments, Paris, France) using the following protocol: 3 cycles of 15 s at 6000 rpm with a 5 s pause. The homogenate was then used in a chloroform-based extraction protocol. RNA pellets were resuspended in 150 µL of DNase/RNase-free water and RNA concentration was measured using a NanoPhotometer NP80 (Implen, München, Germany). All samples were exposed to DNase treatment with a PerfeCTa^®^ Dnase I kit (Quanta Biosciences, Beverly, MA, USA) following manufacturer instructions. Approximately 500 ng of DNase-treated RNA was then reverse transcribed into cDNA using qScript^®^ SuperMix (Quanta Biosciences, USA). RNA extracts were stored at −80 °C and cDNA stored at −20 °C.

Deformed wing virus levels were assessed in a total of 130 individual bees using qPCR. All primer sequences used in qPCR reactions and calculated primer efficiencies are available in Appendix A. Deformed wing virus qPCR primers (DWVQ_F1 and R1) were selected from Martin et al. (2012) and amplified with a 145 base pairs (bp) product within the RNA-dependent RNA polymerase region of DWV. This primer set did not overlap with DWV-dsRNA targeting regions. Two *A. mellifera* genes, Ndufa8 and Pros54, were chosen as validated internal references to normalise DWV levels [48]. Quantitative PCR reactions were validated by checking the primer efficiencies of each target and confirming that efficiencies were close to 2.00 [49]. Samples were analysed in duplicate, and water-only controls (i.e., no DNA/RNA template controls) were included for each target. Each reaction was made to a final volume of 20 µL comprising 10 µL of PowerUp SYBR Green™ Master Mix (Applied Biosystems/Thermo Fisher Scientific, Waltham, MA, USA), forward and reverse primers to a final concentration of 0.6 µM, and cDNA to a final concentration of 1 ng/µL. Reactions were run in 96-well plates on a QuantStudio 7 Flex Real-Time PCR platform (Applied Biosystems/ThermoFisher Scientific, MA, USA) using the following fast cycling conditions: 2 min hold at 50 °C, 2 min hold at 95 °C followed by 40 cycles of 3 s at 95 °C and 30 s at 60 °C; a melt curve was included at the end of each run to check for specificity, and confirmed the amplification of a single PCR product.

Relative expression levels were calculated from cycle threshold (Ct) values of qPCR results using the 2^−ΔΔCt^ method [50]. Raw Ct values are provided in Appendix A. Due to the non-normality and heterogeneity of variance in the data, PERMANOVA was used to test for differences in log-transformed DWV levels (log(2^−ΔΔCt^)) between treatments with an interaction between treatment and parasitism (i.e., whether bees were parasitised or not). PERMANOVA was run using the adonis2 function in the vegan 2.6–4 package [51] in R 4.3.1 (R Core team, Vienna, Austria) with Euclidean distance. We accounted for multiple comparisons between parasitism phenotype using a Dunn’s post hoc test with Benjamini and Hochberg correction to determine which parasitism conditions were significantly different.

### 2.7. RNA-Sequencing of Brood-Parasitising Varroa Mites

RNA was extracted from a total of eighteen pools of seven foundress varroa mites (six pools per dsRNA/control treatment) to investigate treatment effects on gene expression and viral community and abundances in mites. Mites were pooled in 1.5 mL reinforced tubes (Sarstedt, Nümbrecht, Germany) containing approximately 10 × 0.5 mm stainless steel beads and 600 µL of TRIzol^®^ reagent (Life Technologies, Carlsbad, CA, USA). Samples were homogenised in a Precellys^®^ Evolution homogeniser (Bertin Instruments, Montigny-le-Bretonneux, France) at 6800 rpm for 2 cycles of 30 s with a 15 s pause followed by incubation at room temperature for 5 min. Varroa RNA was then extracted using the Direct-zol™ MicroPrep (Zymo Research, Irvine, CA, USA) kit following manufacturer’s instructions with DNase treatment and an additional column RNA wash step of 350 µL prior to the final RNA wash step. Samples were eluted in 15 µL of DNase/RNase-free water and RNA concentration was measured using NanoPhotometer NP80 (Implen, Germany). RNA samples were dried using GentegraRNA™ tubes (Gentegra, Pleasanton, CA, USA) and then sent for sequencing via Custom Science (Auckland, New Zealand). Library preparation included selection of poly(A) tail RNA sequences. Sequencing was conducted on a NovaSeq 6000 platform (Illumina, San Diego, CA, USA) using 150 base pair paired-end sequencing.

Quality of raw RNA-seq reads was assessed using FASTQC 0.11.7. Reads were then cleaned using Trimmomatic 0.39, which were performed under the following conditions: ILLUMINACLIP 2:30:10 SLIDINGWINDOW: 4:20 MINLEN: 25 and read quality was reassessed with FASTQC 0.11.7. The *V. destructor* reference genome and annotation files (GCF_002443255.1_Vdes_3.0) were downloaded from NCBI database https://www.ncbi.nlm.nih.gov/genome/annotation_euk/Varroa_destructor/100/ (accessed on 17 March 2022). We used HISAT2 2.1.0 to index the *V. destructor* reference genome, align clean reads to it and generate a sorted alignment file SAMTOOLS 1.10. Unmapped reads were kept for downstream viral analysis. The resulting BAM files were used for transcript assembly using StringTie 1.3.5 [52] based on the *V. destructor* genome annotation file. Transcripts were then standardized across samples by merging the resulting gtf files to create a non-redundant set of transcripts across all samples. StringTie was run again with expression estimate mode to obtain transcript abundances based on the merged assemblies. Finally, this output was used to generate count matrices for genes and transcripts using prepDE.py script within StringTie.

### 2.8. Differential Gene Expression

An analysis of differentially expressed genes in varroa RNA-seq samples was conducted by first summing transcripts for the same gene annotation and filtering lowly expressed genes from the dataset. Lowly expressed genes were defined as genes with a value of less than one count per million (cpm) for two or fewer samples. Summarising transcripts to the gene annotation level and filtering lowly expressed genes resulted in 10,832 genes to be analysed. Gene counts were normalised using the TMM method from the edgeR R package [53].

For differential gene expression analysis, the voom function in the Limma R package [54] was used to transform gene count data to log2-counts per million and generate mean-variance relationships for each gene. Voom output was used for linear modelling using the lmFit function in Limma with contrasts investigated between non-specific dsRNA vs. DWV-dsRNA, sugar control vs. DWV-dsRNA and sugar control vs. non-specific dsRNA. The plotSA function was used to visualise the fit of the linear model by plotting residual standard deviations for each gene against the average log expression. The treat function in Limma was then used to determine differentially expressed genes between the defined contrasts above, defined by genes with a log2-fold change of 1.10 and *p*-value < 0.05.

A list of 41 RNAi associated genes (Appendix A) was generated based on previous *V. destructor* transcriptome studies [55,56] and used to specifically investigate differences in RNAi responses between samples from different treatment groups. TMM-normalised gene counts for each sample were investigated by multidimensional scaling (MDS) to assess whether samples from the same treatment group had similar expression patterns to the candidate RNAi associated genes using the plotMDS function in Limma package 3.56.2 [54]. Kruskal–Wallis tests were then used to determine whether TMM-normalised gene counts for each RNAi associated gene was significantly different between treatments.

### 2.9. Viral Community Analysis in Varroa Mites

For analysis of viral abundance and community in varroa RNA-seq samples, reads that did not map to the *V. destructor* reference genome were used to assemble transcripts de novo with Trinity 2.10.0 [57]. Trinity transcripts were then aligned using DIAMOND 2.0.7 [58] to the NCBI viral database downloaded on 15/07/2022. Results from the DIAMOND BLAST output was filtered through a custom R script that selected assembled transcripts with a percent identity hit greater than 70 and length greater than 166 amino acid residues, then selected for hits with the best bit score, e-value and percent identity, in this order. Virus names of top hits were retrieved through Entrez [59]. Viral transcript abundances were determined using Salmon run through Trinity (align_and_estimate_abundance.pl script), expressed as transcripts per million (TPM).

Transcripts per million values were first normalised to the filtered, TMM-normalised library size of each sample and non-invertebrate viruses were excluded from the dataset. Differences in virus abundances between treatments were tested using permutational multivariate ANOVA (PERMANOVA) with Bray–Curtis distance calculated between samples using the adonis2 function in the R package vegan 2.6–4 [51]. To specifically test for differences in DWV loads in mite samples between treatments, ANOVA was conducted on log-transformed TPM values of DWV between treatments.

Deformed wing virus sequences assembled from the varroa mite RNA-seq data were investigated further for nucleotide differences in the DWV-dsRNA target regions and to determine the relationship to other DWV variants. Two DWV contigs were assembled by Trinity, both 9223 bp in length and sharing 99.9% pairwise identity, differing by only three nucleotides in the 5′ region. Due to this high sequence similarity only one DWV contig was used for further analysis. The pairwise identity between the Trinity assembled DWV sequence and the three DWV-dsRNA sequences used in our study was analysed by mapping DWV-dsRNA sequences to the partial DWV genome sequence using the software Geneious Prime 2020.2.3 (Biomatters, Auckland, New Zealand). Fifty complete DWV genomes were downloaded from the NCBI database (Appendix A) and a multiple sequence alignment using MUSCLE 3.8.425 [60] was conducted with the full sequence of all 50 DWV sequences and our Trinity DWV contig (DWV_Trinity_DN369_ci_gi_i12). Only DWV-A sequences were used in the alignment. Once aligned DWV sequences were trimmed to remove large gaps resulting in an 8753 bp long alignment. The relationship between our partial Trinity-assembled DWV genome and other DWV genome sequences was assessed with Bayesian phylogenetic inference using a GTR substitution model implemented with MrBayes 3.2.6 [61] plug-in in Geneious Prime 2020.2.3. Bayesian analyses were replicated four times, each with four Markov chains of 2 million generations. Trees were sampled every 2500 generations, of which the first 150,000 generations were discarded as burn-in.

## 3. Results

### 3.1. Deformed Wing Virus Loads in Uncapped Bees

Deformed wing virus loads were determined using qPCR on parasitised individuals with and without wing deformities and non-parasitised individuals from each of the three treatments. There was substantial variation in DWV loads between individuals (Figure 2). Parasitised bees had the highest levels of DWV while most, but not all, non-parasitised bees had low DWV infections (Figure 2). Interestingly, relative DWV load of DWV-dsRNA treated, non-parasitized bees ranged from 2 to 10 log-fold lower than the mean DWV load of non-parasitised bees from the sugar water control (Figure 2). Although PERMANOVA results revealed no significant interaction between treatment and parasitism (F = 2.611, R2 = 0.0540, *p* = 0.07), it is worth noting that this result was just outside the chosen significance threshold (*p* < 0.05). PERMANOVA results showed that DWV levels were not significantly different between treatment groups (F = 0.455, R2 = 0.009, *p* = 0.66), while parasitism had a significant effect on DWV levels in uncapped bees (F = 15.502, R2 = 0.160, *p* < 0.01). A Dunn’s post-hoc test using Benjamini and Hochberg *p*-value adjustment was used to investigate multiple comparisons between parasitism levels to determine which groups were significantly different. Deformed wing virus levels were significantly lower in non-parasitised bees compared to parasitised bees with deformed (Z = −4.604, *p*-adj < 0.001) or normal wings (Z = −2.102, *p*-adj = 0.03). Deformed wing virus loads were also significantly different between parasitised bees with normal and deformed wings, with deformed individuals having significantly higher virus levels (Z = 2.791, *p*-adj < 0.01).

### 3.2. Instance of Wing Deformities in Uncapped Bees

All brood cells were uncapped from each mini-frame to obtain the total number of bees with wing deformities for each mini-hive. Within our experiment, wing deformities were only observed among bees parasitised by varroa. The highest proportion of wing deformities among uncapped bees from a mini-hive was 12% (Figure 3). ANOVA results showed the proportion of wing deformities among uncapped bees was not significantly different between treatments (df = 2, chi-squared = 1.4251, *p* = 0.49).

### 3.3. Differential Gene Expression in Varroa Mites

Differential gene expression analysis found that, of the 10,832 genes in the dataset, none were significantly differentially expressed between treatments using the criteria of a log2 fold change of 1.10 and a significance threshold of 0.05. Consistent with this result, no obvious trend was apparent in the plotted results of mean–variance relationships for each gene or visualization of the linear model fit (Appendix A).

Forty-one RNAi-associated genes were selected as candidate genes from the literature for further investigation (Appendix A). TMM normalised gene counts of these candidate genes were used to perform multidimensional scaling (MDS) analysis to explore if treatment groups had similar expression patterns of these 41 RNAi-associated genes. Plotting the first two dimensions of the MDS accounted for 92% of the variation in the data (Figure 4). No clustering of samples from the same treatment group was observed.

Differences in gene expression (TMM normalised gene counts) for each candidate gene were assessed using Kruskal–Wallis tests. Consistent with results from the differential gene expression analysis, expression of these 41 candidate genes were not significantly different between treatments (*p* > 0.05) (Appendix A).

### 3.4. Viral Community and Abundances in Varroa RNA-Seq Samples

Virus abundances in varroa samples were not significantly different between treatments (F= 1.8758, R2 = 0.2001, *p* = 0.15) based on PERMANOVA results using Bray–Curtis distances. Deformed wing virus loads across treatments are shown in Figure 5, and ANOVA showed that loads were not significantly different between treatments (df = 2, F = 2.883, *p* = 0.09). Five invertebrate-associated viruses were identified in RNA-seq results of varroa mite pools. These viruses included three common honey bee-associated viruses: DWV-A (hereafter DWV), Black queen cell virus (BQCV) and Sacbrood virus (SBV). Two varroa specific viruses, varroa destructor virus 2 (VDV-2) and varroa destructor virus 3 (VDV-3), were also found. No transcripts in any of the RNA-seq samples were identified as belonging to DWV-B via DIAMOND BLAST; all DWV transcripts blast to DWV-A genomes. Deformed wing virus was the most abundant virus across most samples (Appendix A), contributing to over 90% of total viral loads in all samples except three. Two of these three samples belonged to sugar water treated mini-hives and DWV made up 28.4% and 40% of total virus loads. The third sample was from a DWV-dsRNA-treated mini-hive and DWV made up 85.1% of total virus loads. Interestingly, SBV was also abundant in these three samples, contributing to 71.1%, 59.7% and 14.7% of total viral loads, respectively. Black queen cell virus was found in 15 samples, making it the only virus not found in all 18 samples, contributing to <0.04% of all viral loads in samples where it was found. Varroa destructor virus 2 ranged between 0.2% and 3.7% of total viral loads. Varroa destructor virus 3 was the least abundant virus, making up <0.01% of total viral loads in all samples.

Two DWV contigs 9223 bp long were assembled using Trinity. The two sequences showed 99.9% pairwise identity, differing by only three nucleotides at the 5′ end. To investigate the pairwise identity between these DWV sequences identified in our mite samples and the DWV-dsRNA sequences used in the experiment, the three DWV-dsRNA sequences were mapped to one of the DWV Trinity sequences. Results showed 98.6%, 97.7% and 98% pairwise identity between DWV-dsRNA 1, DWV-dsRNA 2 and DWV-dsRNA 3, respectively, and the DWV partial genome sequence. These pairwise identities equated to 10 single nucleotide polymorphisms across each DWV-dsRNA region. The pairwise alignment can be found in Appendix A. Phylogenetic analysis showed the DWV Trinity contig (DWV_Trinity_DN369_ci_gi_i12) clustered with other DWV genomes identified in New Zealand *A. mellifera* samples (MN538208.1 and MF623172.1) with strong support (posterior probability of 1) (Figure 6). Two other DWV genomes (MT096518.1 and MT096529.1) also grouped with New Zealand DWV genomes, these sequences were assembled from a metagenomic analysis of honey-baited FTA cards used to sample for viruses in mosquitoes in Spain [62]. According to our phylogenetic analysis, DWV variants in New Zealand are closely related to each other and form a separate clade from USA variants, such as AY292384.1. The DWV Trinity contig (DWV_Trinity_DN369_ci_gi_i12) assembled from our RNA-seq samples and used in this phylogenetic analysis has been deposited into NCBI Genbank with accession ID OR786472.

## 4. Discussion

Several previous studies have shown that RNAi can effectively reduce the development of viral infection in honey bees for a range of pathogens including DWV, SBV and IAPV [35,37,38,39,40]. Recent field trials feeding SBV-specific dsRNA to *A. cerana* colonies found that dsRNA treatment could reduce SBV loads and protect colonies from the development of SBV disease in infected colonies [44]. These field trials also alluded to the possible limits of pathogen-specific RNAi: colonies already showing high SBV loads and SBV disease symptoms were unable to be rescued by SBV-specific dsRNA treatment [44]. Notably, this field study tested the efficiency of SBV RNAi in the context of naturally acquired SBV colony infection. Most pathogen RNAi studies have tested the effectiveness of pathogen-specific dsRNA in individuals orally inoculated with virus after dsRNA treatment [38,40,41,43]. Given the role of varroa as a vector for viral pathogens in honey bees, especially DWV, it is important to investigate the effectiveness of pathogen-specific RNAi in the presence of varroa parasitism. It is highly likely that viral transmission route may play a role in pathogen-specific RNAi efficiency. Additionally, the continuous feeding on honey bee body fat by varroa likely means that single injections of DWV inoculum to bees is not representative of the transmission dynamics of DWV.

Mini-hive experimental designs are a useful tool for assessing effectiveness of RNAi-based technology within a controlled, contained environment that resembles the dynamics within a hive [34,45]. In this study we employed a mini-hive experimental design to test the effectiveness of using DWV-dsRNA to lower viral loads and DWV symptoms in reared larvae. Mini-hives were populated with bees, larvae and varroa with naturally acquired DWV infections. Introducing varroa mites into each mini-hive and measuring viral loads of bees parasitised and non-parasitised during development allowed the effects of DWV-dsRNA on viral loads to be compared within the context of an important vector of the virus.

Results from our experiment showed parasitism explaining a significant amount of the variation in DWV virus loads between individuals, a result consistent with previous studies showing the effects of varroa parasitism on virus loads in bees [1,16,63]. Individuals with deformed wings had the highest viral loads while virus loads in non-parasitised bees were significantly lower than parasitised individuals with either deformed or normal wings. However, our experiment did not find a significant difference in DWV loads between uncapped bees from different treatments. Consistent with this finding, DWV-dsRNA did not reduce the proportion of bees with wing deformities. Our results also showed no significant interaction between treatment and parasitism phenotype, indicating that the effects of DWV-dsRNA treatment did not depend on parasitism phenotype. This interaction was just outside our significance threshold of 0.05. Interestingly, for non-parasitised bees the average DWV load of individuals from DWV-dsRNA treated mini-hives was 2 orders of magnitude lower compared to the average DWV loads of individuals from sugar control treated mini-hives (Figure 2). We hypothesize that the effects of DWV-dsRNA on viral loads in developing bees could be impacted by their parasitism phenotype and warrants further investigation.

Although the concentration of dsRNA administered to mini-hives was consistent with a previous mini-hive RNAi study [34] it is possible that higher concentrations of DWV-targeting dsRNA could have resulted in a significant reduction in DWV loads among uncapped bees. Specifically, it is difficult to know the final concentration of dsRNA deposited into brood cells by nurse bees after foraging on the DWV-dsRNA. While this transmission route has been shown by previous studies to elicit RNAi in bees [34,45], it is possible that higher concentrations of administered dsRNA in this study were needed for enough dsRNA to be transferred to the brood and lower DWV loads in developing bees. Desai et al. (2012) found rearing adult bees and larvae on DWV-dsRNA directly, prior to oral inoculation with DWV, resulted in a 300-fold reduction in DWV levels compared to bees only fed virus or fed virus and a non-specific dsRNA. Oral inoculation with DWV in the Desai et al. study was intended to be enough for eliciting DWV symptoms, resulting in high DWV levels that may not be representative of levels acquired naturally through oral transmission from nurse bees or other oral routes, which tends to result in asymptomatic infection [16,63,64]. In contrast to Desai et al. (2012), bees in our experiment were not orally inoculated with DWV. Instead, our experiment took advantage of varroa’s role as a vector for DWV to compare the effectiveness of DWV-dsRNA to reduce DWV loads in parasitised and non-parasitised bees. Therefore, differences between our findings may be due to this difference in DWV infection route (i.e., oral inoculation with DWV inoculum vs. varroa-vectored DWV). More recently, results from Leonard et al. (2020) have shown that inoculating newly emerged adult bees with symbiotic bacteria expressing DWV-dsRNA significantly reduced virus loads and increased survival in bees experimentally infected with DWV. This important difference in DWV-dsRNA delivery method between Leonard et al. (2020) and this study likely contributes to our differences in results and generally makes drawing comparisons between our studies difficult. We believe our experimental design tested the effects of DWV-dsRNA on viral loads that more closely resembled those experienced by bees within a colony compared to experimentally induced DWV infection used by previous RNAi studies.

RNA interference is a highly sequence-specific mechanism, and interestingly previous pathogen-specific RNAi research has shown that as little as 5% sequence divergence between a pathogen-specific dsRNA sequence and the complementary viral genome region may be enough to significantly reduce the effectiveness of RNAi [65]. Therefore, it is possible that the diversity of DWV variants infecting New Zealand bees and varroa is unique [66] and the dsRNA sequences used in this study are not representative of the variants circulating among New Zealand bees and varroa. Our phylogenetic analysis showed that the DWV sequence assembled from the mite RNA-seq samples in this study is closely related to other DWV variants in New Zealand and these sequences form a monophyletic group with two other DWV genomes assembled from metagenomic data of honey-baited FTA cards containing mosquitos [62] (Figure 6). Interestingly, the authors of that study acknowledge the DWV sequences could be derived from the honey-baited cards themselves rather than the mosquitoes [62]. Overall, our phylogenetic analysis is consistent with previous work that has shown that New Zealand DWV variants sampled from a range of hosts are closely related and appear distinct from USA DWV lineages [66]. Mapping the three DWV-dsRNA sequences to the partial DWV genome assembled from our RNA-seq mite samples showed between 1.4% and 2.3% sequence divergence at each of the DWV-dsRNA target regions. Although this level of sequence divergence is less than the 5% sequence divergence identified in a previous study [65], it is possible this level of difference could influence the efficiency of each sequence to knock down DWV loads. We propose that sequence divergence between pathogen-variant and pathogen-specific dsRNA is highly important for future pathogen-specific RNAi studies to consider, especially those interested in viral pathogen control.

Targeting DWV in varroa could be helpful for reducing the negative effects of DWV in honey bee colonies, therefore, we used RNA-seq to assess the effects of DWV-dsRNA exposure on viral loads and gene expression in varroa mites. RNA-sequencing results did not show any differences in viral abundance or viral communities between varroa samples from different treatments. Deformed wing virus dominated the viral community in most samples, illustrating that exposure to DWV-dsRNA either through contact with brood food and/or from host feeding did not knockdown DWV loads in mites (Figure 5). Recently, we investigated viral communities in bees and varroa from the same hives located around New Zealand and found that DWV is highly prevalent and can be highly abundant in mites [67]. Interestingly, results from Lester et al. (2022) [67] observed a significant correlation between high loads of VDV-2 in mites and low loads of DWV in bees collected from the same hive. This pattern was not observed in this experiment, possibly due to the low abundance of VDV-2 across all samples analysed in our RNA-seq dataset. However, the presence of VDV-2 across all mite samples is consistent with results from Lester et al. (2022) and others [68,69,70] which have demonstrated VDV-2 is highly prevalent among varroa mites.

Differential gene expression analysis did not show any significantly differentially expressed genes between mite samples from different treatments. To specifically investigate whether mites from dsRNA treated mini-hives had an elevated RNAi response, further analyses of specific RNAi associated genes identified in the literature [55,56] were conducted. Consistent with the differential gene expression analysis results, expression of none of these genes was significantly different between treatments.

Given the relatively long period between sampling of bees and mites and initial treatment (approximately 16 days), the dynamics of DWV levels in pupae and parasitising mites early in the experiment are unknown. Any initial change in DWV levels in response to treatment may not be reflected in these results and may have been negated by a waning treatment effect during pupation combined with continued DWV transmission in parasitised individuals. Likewise, it is possible that any RNAi response to dsRNA treatment may have waned by the time mites were sampled approximately 12 days after initial exposure. Future research could address this limitation by daily sampling of parasitised larvae post cell capping to track how DWV infection develops in parasitised, dsRNA treated individuals. Standardizing viral loads across mites and bees would be an important point of control in such an experiment as individual mites can vary dramatically in their vector competence [71,72].

Next generation biopesticides offer promising control strategies for combatting pests and pathogens and cost-efficient production of such technology will bring products to market [41,73,74]. The importance of honey bees to global food security combined with the range of pests and pathogens they face suggests the apicultural industry may especially benefit from these technologies [35,38,44,75]. However, we argue that when investigating RNAi-technologies aimed at controlling viral pathogens it is important to consider how different transmission routes of viral pathogens may impact their efficiency. Transmission route has proven to be highly important in disease outcome and infection loads of DWV, especially when transmitted by varroa [61,76]. For colonies experiencing SBV disease symptoms, research has shown that RNAi was insufficient for rescuing the colony from death, even when SBV-specific dsRNA was administered at high concentrations [44]. Similarly, it is possible that the amount of DWV vectored by varroa to bees, combined with the negative effects of varroa parasitism on bee immune system [77,78], overwhelms the ability of the RNAi system to reduce DWV levels.

## Figures and Tables

**Figure 1 viruses-15-02259-f001:**
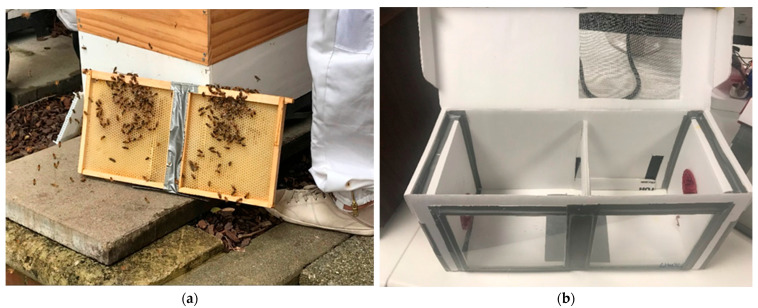
A mini-frame and mini-hive used in experiments. (**a**) Two mini-frames adhered together to fit as a single frame inside a hive for acquiring brood. Mini-frames were separated for one mini-frame of larvae to be introduced to the mini-hive. (**b**) A mini-hive with two plexiglass viewing windows in the front, and two chambers separated by a plexiglass partition to fit a mini-frame in the left chamber and the treatment pouch in the right foraging chamber.

**Figure 2 viruses-15-02259-f002:**
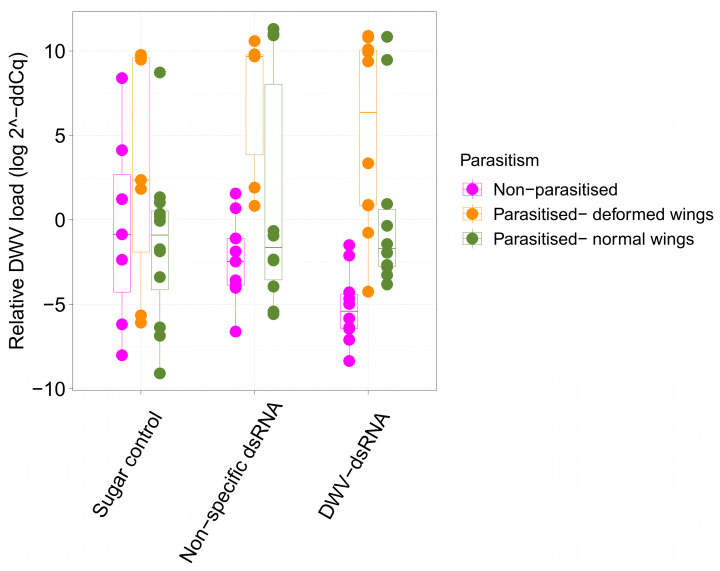
Relative DWV levels of uncapped bees from sugar, non-specific dsRNA and DWV-dsRNA treated mini-hives. Coloured boxes show different parasitism phenotypes: non-parasitised (pink), parasitised with wing deformities (orange) and parasitised with normal wings (green). Raw data points (i.e., Relative DWV level of individual bees) for each box plot are overlayed. Upper and lower hinges of the boxes show 75% and 25% quantiles, respectively, separated by black lines showing the median. Upper and lower whiskers extend to 1.5* interquartile range.

**Figure 3 viruses-15-02259-f003:**
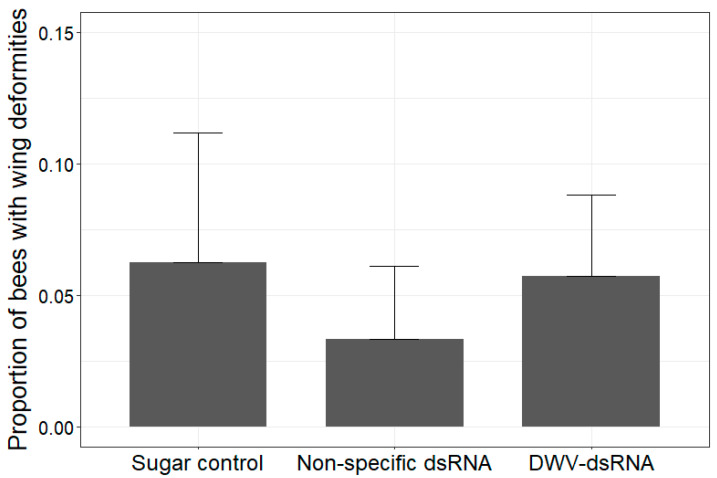
Effect of treatment on the proportion of uncapped bees with wing deformities from each mini-hive. Bars show mean proportion of uncapped bees with wing deformities for sugar water (n = 4), non-specific dsRNA (n = 4), and DWV-dsRNA (n = 5) treated mini-hives with error bars showing standard deviation. No significant difference in proportion of bees with deformed wings was found between treatments (df = 2, chi-squared = 1.4251, *p* = 0.49).

**Figure 4 viruses-15-02259-f004:**
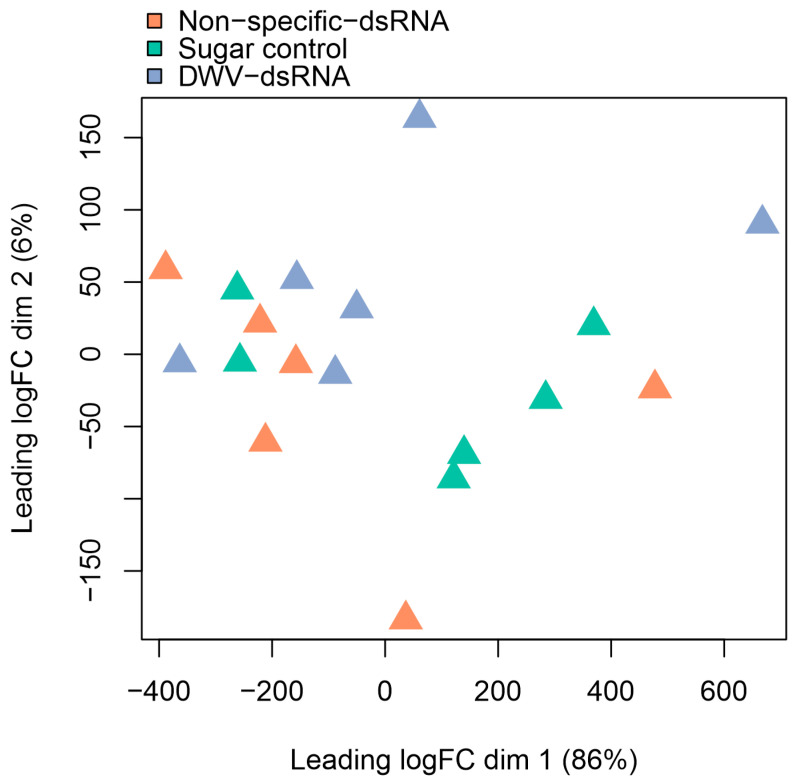
Multidimensional scaling (MDS) plot of expression profile (TMM normalised gene counts) of candidate RNAi-associated genes for each mite sample. Each triangle denotes an individual mite sample with colours indicating which treatment group the sample belongs to: Non-specific dsRNA (orange), sugar control (green) or DWV-dsRNA (purple). Dimensions one and two are shown and account for 92% of the variation in the data.

**Figure 5 viruses-15-02259-f005:**
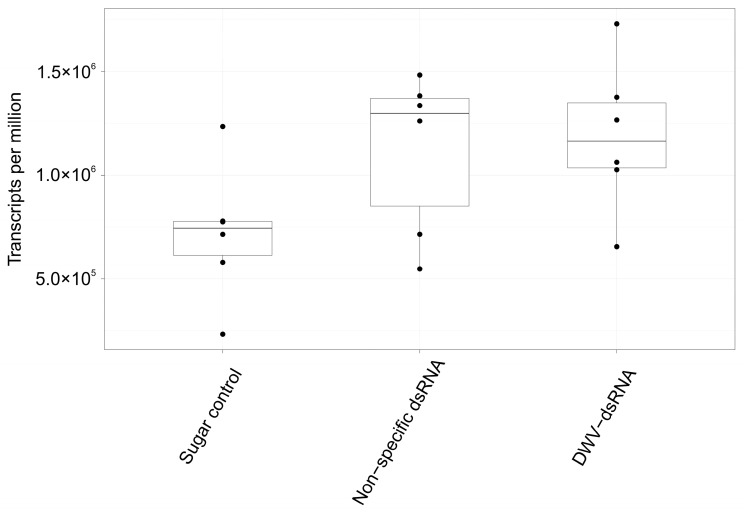
Relative abundance of deformed wing virus loads represented as transcripts per million (TPM) in varroa mite RNA-seq samples from sugar water, non-specific dsRNA and DWV-dsRNA-treated mini-hives. Upper and lower hinges of the boxes show 75% and 25% quantiles, respectively. Whiskers extend to 1.5* interquartile range. Raw data is overlaid to show TPM values for each sample (black points). ANOVA results showed no significant difference in DWV loads in varroa mites between treatment groups (df = 2, F = 2.883, *p* = 0.09).

**Figure 6 viruses-15-02259-f006:**
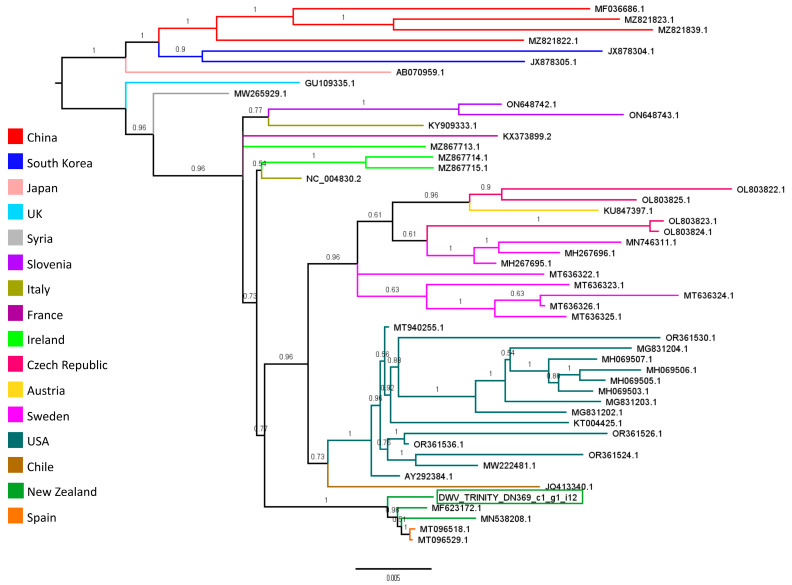
Unrooted Bayesian phylogenetic tree of DWV-A genome sequences, labelled with NCBI accession IDs. Branch labels show posterior probabilities and are coloured by country of isolate sample origin. The DWV contig identified in varroa samples from our study is shown in the green box.

## Data Availability

Raw FastQ files for each RNA-seq sample has been submitted to the NCBI Sequence Read Archive under BioProject PRJNA1036652 with accession numbers SAMN38145358-SAMN38145375. The DWV contig assembled from RNA-seq samples (DWV_Trinity_DN369_ci_gi_i12) has been deposited into NCBI Genbank with accession ID OR786472.

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
