# Peer review of "Effects of Deformed Wing Virus-Targeting dsRNA on Viral Loads in Bees Parasitised and Non-Parasitised by Varroa destructor"

_viruses, 2023, doi:10.3390/v15112259_

Round 1

Reviewer 1 Report

Comments and Suggestions for Authors

The authors set out to investigate the ability of DWV-dsRNA treatments to reduce DWV loads in bees. Using a mini-hive design and inoculating hives with mites they aimed to create a natural hive condition, with existing virus levels interacting with mite parasitism rather than oral inoculation of bees with virus. Unfortunately, qPCR and RNAseq analysis found no significant effect of treatment on DWV levels.

Overall, the authors have presented a well-written study that clearly articulates the value of their study, describes an elegant mini-hive design and provides reasonable explanation for the lack of response to DWV-dsRNA treatment. However, I’d like the authors to address some concerns with the dsRNA treatment in their study. 

1. There was no initial confirmation of efficacy for the dsRNA treatment. A simple feeding experiment of the dsRNA treatment would provide confidence that the dsRNA was viable and could also be used to identify a useful concentration for the mini-hive experiment. Do the authors have any data to confirm the viability of the dsRNA synthesised?

2. There needs to be more discussion of how the dsRNA treatment is expected to reach larvae to reduce virus levels. The dsRNA treatment is being consumed by worker bees who then need to transfer this to larvae via the brood food. This is certainly possible but also a long pathway that will influence the final efficacy of the treatment. This also comes back to the assumptions of the efficacy of the dsRNA and concentration used. Was enough viable dsRNA getting to the brood to affect virus levels?

3. The authors present virus sequence variation as a potential factor affecting treatment efficacy. Surely this was considered in the design of the dsRNA? Can they use their RNAseq data from this or previous work to test this and compare the dsRNA sequence with the DWV sequence variants observed in the hives?

4. Lastly, the final concluding statements feel contradictory to earlier points in the discussion. It says that the amount of DWV vectored by mites may have overwhelmed the treatment, but earlier it is suggested that DWV-A is not well vectored by varroa.

Author Response

We appreciate Reviewer #1 taking the time to read and review our manuscript. We found their comments helpful for us to expand some of our analyses and clarify points in our discussion. Below we address each of their comments and briefly discuss what changes we have made relating to this feedback. 

  1. The suggestion to have used an initial feeding experiment to confirm the efficacy and determine a concentration for the mini-hives in our study is a useful suggestion. Unfortunately, no initial feeding experiment was conducted.
  2. We agree with the reviewer’s comments that the transmission of dsRNA from adult bees to larvae through consumption and deposition in brood cells as larval food is a long transmission pathway and it is very difficult to understand the final concentration of dsRNA reaching the larvae. Moreover, there may be variation in exposure to dsRNA between larvae within the same mini-hive given this transmission route. However, we would like to highlight that Garbian et al. 2012 and Maori et al. 2019 both showed feeding 200-300 ug of dsRNA per feeding to mini-hives of 250 bees and larvae resulted in effective gene knockdown in parasitising varroa or bees. In our study, DWV-dsRNA treated mini-hives were fed a total amount of 1,800 ug of dsRNA per feeding (30 ml of dsRNA sugar water at a concentration of 0.06 mg/ml). Therefore, considering methods published by previous groups we assumed this concentration would be more than enough to impact virus loads in bees although this did not appear to be the case, possibly due to insuffient levels reaching larvae in our experimental design or dissimilarity between dsRNA and DWV variants (which we discuss later). While we acknowledged the limitation of dsRNA concentration in the discussion, we have added to this section to more clearly describe how this relates to the transmission route of the dsRNA and refer to the concentration fed to mini-hives possibly being insufficient for enough dsRNA transfer to larvae from nurse bees (lines 530-534).
  3. We have conducted additional analyses to address the reviewer’s comment to further investigate our hypothesis of dissimilarity between DWV variants in NZ bees and mites and the DWV-dsRNA sequences used in the study as a possible explanation for the lack of DWV knockdown. We assembled two DWV contigs from our mite RNA-seq samples with 99.9% sequence identity. The three DWV-dsRNA sequences used in this study were mapped to one of these assembled contigs and we report the pairwise identity for each of the DWV-dsRNA regions (lines 467-469) to show differences between the DWV variants present in our samples and the DWV-dsRNA sequences used in the study. The pairwise comparisons are also provided in supplementary File S3. 
  4. We understand how our final conclusions may have appeared contrary to earlier points in the discussion on the ability of DWV-A to replicate in varroa mites. These earlier points have been removed in the revised manuscript. 

Reviewer 2 Report

Comments and Suggestions for Authors

This was a very nice study with a tight and well executed design. There may be some room for improvement in the analysis. I will say that acceptance of the manuscript should be delayed until the authors show that their raw reads have been submitted to NCBI's SRA.

1. Add a few lines at the end of the introduction about what you DID find.

2. Why did the authors choose to provide a much higher dose of the non specific dsRNA? The heatmap in figure 5 does seem to suggest these mites look more like mites from sugar fed control mini-colonies but it does beg the question whether these mites are a true control for host exposure to dsRNA. Can the authors comment on this and perhaps address it in the Discussion?

3. Fig 2 – I think you mean log (2^-ddCq)

4. You performed a de novo assembly for the viral community analysis in varroa. This provides a really nice opportunity to compare the DWV your bees were exposed to to published sequences and the primers you used for qPCR. I think you should check your primers against the full consensus DWV contig from your mite samples to make sure they match and perhaps do a simple comparison to published sequences. Ideally a small maximum likelihood phylogeny showing the relationship to other published sequences.

5. What were the cq values for DWV? It's not clear from the data how DWV loads relate to other similar studies. Since no absolute quantification was done it will never be perfectly comparable anyway, but Cq values give at least some idea. In fact, please provide raw data in supplementary tables including the reference gene Cq values and the primer efficiencies that were calculated. Primer efficiencies can also simply be included in the methods.

6. The use of FPKM for the chosen subset of genes is odd and not recommend for between sample comparisons due to inconsistent FPKM normalized library count sizes. Instead, you should use the DESeq2 or Edge R normalization methods and repeat the Kruskal-wallis tests for the chosen subset of genes.

See this resource: https://hbctraining.github.io/DGE_workshop/lessons/02_DGE_count_normalization.html

7. You reference Fig S2 in line 465 but I don’t see a Fig S2 in the downloaded files.

8. Did any of your reads from varroa align to the MPK4a soybean (Glycine max) gene? This would indicate if any if your mites are consuming your provided treatment. This may also indicate an inflation of the TPM values in figure 6 in mites from DWV dsRNA fed bees. Assuming, of course, your treatment was transferred to developing larvae or your mites may have acquired some of your treatment during "phoresy" on nurses.

Author Response

We would like to thank reviewer #2 for taking their time to review and provide helpful feedback on our manuscript, which we believe has helped improve our paper. Specifically we found their advice to use TMM normalised read counts rather than FPKM normalisation and repeat our Kruskal-Wallis tests particularly helpful for this, as well as our future, work. Additionally, their suggestion to include a phylogenetic analysis of the assembled DWV sequences in our RNA-seq samples was useful for expanding on our results and hypotheses. Below we address each of the reviewer's comments in more detail and provide a brief description of the manuscript changes.

  1. Our RNA-seq raw reads have been submitted to the NCBI SRA database (BioProject PRJNA1036652: accession numbers SAMN38145358- SAMN38145375). 
  2. We appreciate the reviewers suggestion to add some lines about our findings at the end of our introduction however, the final lines of our last paragraph of the introduction clearly present the hypotheses of our study and we believe lead well into the rest of our study. We didn’t find it appropriate to add information in this section on what our study found as this is typically resevered for the results and discussion. A brief summary of our results and findings are provided in the abstract.
  3. The non-specific dsRNA used in this study was already available to us in the laboratory from a previous experiment and we believed it would be useful for this study as a non-specific control. While we agree with the reviewer that diluting the dsRNA to a similar concentration as the DWV-dsRNA treatment may have been a more robust study design, our results showed no significant difference in gene expression or viral load between non-specific dsRNA and sugar control treated mites.
  4. We have adjusted the y-axis legend in Figure 2.
  5. We have addressed the reviewer’s comment to double check that the qPCR primers used in our study match the assembled DWV contig in our RNA-seq samples. The partial DWV genomes were assembled in our RNA-seq samples and had 99.9% pairwise similarity, differing only in the 5’ region of the genome. The qPCR primers used in our study were checked in Geneious against one of these partial DWV genomes and both forward and reverse primers were found to bind to our DWV contig. Additionally, we conducted a phylogenetic analysis, per the reviewer’s suggestion, to assess how closely related the partial DWV genome sequence identified in our mite RNA-seq samples is to other DWV genomes. We have included this bayesian inference phylogenetic tree in our manuscript (Figure 6) as well as reported the pairwise identities between the DWV-dsRNA sequences and their respective regions of the DWV partial genome assembled in this study.
  6. As per the reviewer’s request, we have provided the raw qPCR data in a supplementary table (Table S1). However, we have to disagree with the reviewer’s assessment that Ct values are very comparable across different studies conducted by different research groups from different laboratories with different protocols as well as different instruments used in these laboratories for conducting qPCR. Additionally, primer sequences and efficiencies are provided in File S1, Table S1 (line 251).
  7. We appreciated this critique of our analyses from the reviewer. We have repeated the Kruskal-Wallis tests for our set of candidate genes using TMM normalised counts for each gene. TMM normalisation was conducted using the Edge R package. We have updated the manuscript to reflect this change in methods and results.
  8. We have added FigureS2 to the supplementary material.
  9. The RNA sequencing performed in this study included a poly-A selection step in the library prep which we had forgotten to acknowledge in the methods. We have clarified this in the methods in line 290. Due to this selection step it’s unlikely the MPK4a soybean gene would have been sequenced during RNA-seq. Likewise, it’s unlikely our RNA-seq also picked up DWV-dsRNA in our samples, which indeed would have skewed our TPM relative expression of DWV loads in mite samples.

Reviewer 3 Report

Comments and Suggestions for Authors

Reviewer #

Viruses - 2673766

Authors: Smeele, Baty, and Lester

Title: Effects of Deformed wing virus-targeting dsRNA on viral loads in bees parasitised and non-parasitised by Varroa destructor

This well-written manuscript reports on the evaluation of double-stranded RNA (dsRNA) administration on the overall loads of Deformed Wing Virus (DWV) in DWV-infected host honey bees (Apis mellifera).  Multiple studies have been conducted by others on the use of RNA silencing (RNAi) to lower infection levels of DWV and other RNA viruses in bees.  Those previous studies have indicated that it is possible to lower the level of infection in honey bees by feeding virus-specific dsRNA to bee larvae.  However, this report is the first to evaluate an anti-virus RNAi strategy in the presence of the ectoparasite Varroa destructor, while it feeds on the honey bees, and in doing so actively transmits DWV to the parasitized bees.  As such, Varroa is a major problem for bee colony survival.

This study uses a novel strategy involving constructed mini-hives to test the effectiveness of dsRNA treatment of bees with and without Varroa on reducing the level of DWV levels in those bees.  The experimental results are well controlled and statistically sound.  The authors discovered that different treatments had no significant difference in DWV loads in bees.  Likewise, DWV-dsRNA did not reduce the proportion of bees with wing deformities, which is a pathogenic outcome of DWV infection of Apis mellifera.  To their credit, the authors also showed that for non-parasitized bees the average DWV load of individuals from DWV-dsRNA treated bees was about 100-fold lower compared to average DWV loads in bees of control mini-hives.  This result indicated that the choice of dsRNA specific to DWV was effective in reducing virus levels, probably through the RNAi pathway. 

Overall, this manuscript describes a well-planned and thorough study that shows that different transmission routes of viral pathogens may impact the relative efficiency of anti-viral RNAi strategies in bee colonies.  Thus, the DWV-transmission route, especially when transmitted by varroa parasites, must be considered in treatment regimens.  This study provides a significant advance in the control of DWV infection of honey bees.

Prior to publication, this reviewer asks the authors provide (Table form) the sequences of the DWV dsRNAs used for RNAi treatments here.  Where in the DWV genome are these sequences found?  The reader should not have to search other publications for these important reagents.  Such information is important in subsequent efforts by others to confirm the results of this study.

Author Response

We would like to thank reviewer #3 for taking the time to read and provide comments on our manuscript. We were glad to read they found our study a significant contribution toward the control of DWV infection in honey bees. This reviewer requested we provide the DWV-dsRNA sequences used in our study. We have included FileS2 in the supplementary material which includes a Table with details on each DWV-dsRNA sequence used in our mini-hive experiment including the sequence itself. We also include in this file a visual representation of the DWV genome regions each sequence was designed to target. This file is now referenced in the methods section lines 197-199.